# Phage tRNAs evade tRNA-targeting host defenses through anticodon loop mutations

**Daan F van den Berg[1,2], Baltus A van der Steen[1,2], Ana Rita Costa[1,2,3], Stan JJ Brouns[1,2,3]\***

[1]Department of Bionanoscience, Delft University of Technology, Delft, Netherlands; [2]Kavli Institute of Nanoscience, Delft, Netherlands; [3]Fagenbank, Delft, Netherlands

**Abstract** Transfer RNAs (tRNAs) in bacteriophage genomes are widespread across bacterial host genera, but their exact function has remained unclear for more than 50 years. Several hypotheses have been proposed, and the most widely accepted one is codon compensation, which suggests that phages encode tRNAs that supplement codons that are less frequently used by the host. Here, we combine several observations and propose a new hypothesis that phage-encoded tRNAs counteract the tRNA-depleting strategies of the host using enzymes such as VapC, PrrC, Colicin D, and Colicin E5 to defend from viral infection. Based on mutational patterns of anticodon loops of tRNAs encoded by phages, we predict that these tRNAs are insensitive to host tRNAses. For phage-encoded tRNAs targeted in the anticodon itself, we observe that phages typically avoid encoding these tRNAs, further supporting the hypothesis that phage tRNAs are selected to be insensitive to host anticodon nucleases. Altogether, our results support the hypothesis that phage-encoded tRNAs have evolved to be insensitive to host anticodon nucleases.

## Editor's evaluation

This important work substantially advances our understanding of the mechanisms phages use to evade host defenses. Specifically, the authors use computational and theoretical analyses of tRNA-encoding phages that infect several bacterial species to identify a novel, potential mechanism through which phage-encoded tRNAs help these phages evade tRNA cleavage that is induced as a host defense. Although the evidence supporting the conclusions is compelling, with multiple observations suggesting that the phage-encoded tRNAs have evolved to evade host-encoded tRNases, the conclusions would have been more strongly supported by providing an experimental test of the hypothesis.

## Introduction

Transfer RNAs (tRNAs) were first discovered in the 1950s (*Kresge et al., 2005*) and have since been found to play a vital role in the central dogma of molecular biology in all living systems (*Crick, 1970*). During the 1960s, tRNAs were also reported in viruses of bacteria (bacteriophages or phages) (*Weiss et al., 1968*). We now know that phage-encoded tRNAs are widespread, especially among virulent phages (*Bailly-Bechet et al., 2007*). Multiple hypotheses have been proposed for the role of these phage-encoded tRNAs. The most established being codon compensation, where codons rarely used by the host but necessary to the phage are supplemented by the tRNAs encoded by the phage (*Bailly-Bechet et al., 2007*). Why phages are pushed toward these alternative codons is generally believed to be a side effect of differences in the GC content of phage and host (*Bailly-Bechet et al.,*

*For correspondence:
stanbrouns@gmail.com

**Competing interest:** The authors declare that no competing interests exist.

*2007*; *Lucks et al., 2008*; *Limor-Waisberg et al., 2011*). A recent study by *Yang et al., 2021* may have hinted at an additional factor: phage tRNAs represent a means to counteract the depletion of host tRNAs that occurs as an early response to phage infection. The host employs several mechanisms to deplete its tRNA pool, such as downregulating the expression of its tRNAs, modifying tRNAs to make them unusable for translation, and most notably cleaving the tRNAs using anticodon nucleases (*Thompson and Parker, 2009*; *Yang et al., 2021*; *Bailly-Bechet et al., 2007*; *Amitsur et al., 1989*; *Wolfram-Schauerte et al., 2022*). Exactly what activates host tRNA cleavage is often unknown; an exception is anticodon nuclease PrrC, which cleaves tRNA-Lys(ttt) when triggered after sensing phage T4-encoded protein Stp (*Kaufmann, 2011*). In response, phage T4 encodes a tRNA ligase that repairs the cleaved tRNA-Lys (*Kaufmann, 2011*). Recently, a phage-encoded tRNA was found to replenish the depleted host tRNA caused by phage defense system Retron Ec78 , thereby preventing the inhibition of phage propagation (*Azam et al., 2023*). However, it remains unclear how phage tRNAs avoid being degraded by the same mechanism that results in the depletion of host tRNAs during phage infection. Here, we have analyzed phage-encoded tRNAs and hypothesize that the tRNAs encoded by phages are insensitive to tRNA anticodon nuclease activity, preventing depletion of the tRNA pool and translation stalling during phage infection.

## Results and discussion

To investigate our hypothesis that phage tRNAs are insensitive to tRNA anticodon nucleases, we analyzed the tRNAs encoded by a large and well-characterized dataset of tRNA-rich bacteriophages (33 tRNAs per phage on average) that infect mycobacteria: mycobacteriophage cluster C1 (*Russell and Hatfull, 2017*; *Figure 1A and B*). We specifically selected this mycobacteriophage dataset because the bacterial host encodes a range of well-characterized tRNA nucleases (tRNAses), such as VapC, MazF, and RelE (*Winther et al., 2016*; *Chauhan et al., 2022*; *Cruz et al., 2015*; *Cintrón et al., 2019*; *Barth et al., 2021*; *Pedersen et al., 2003*). A subset of these tRNAses target the tRNA anticodon loop and are activated upon various stress responses, including phage infection (*Calcuttawala et al., 2022*). Upon activation, these anticodon nucleases cleave specific tRNAs in conserved regions within the anticodon loop to inactivate the tRNAs and thereby regulate protein translation of the host (*Winther et al., 2016*). The cleavage region within the tRNA anticodon loop is sequence-dependent and highly specific to the type of tRNA. Mutations in the recognition and cleavage sequence within the anticodon loop have been found to result in insensitivity to these anticodon nucleases (*Winther et al., 2016*; *Cruz et al., 2015*). To check phage tRNAs for mutations that are known to cause insensitivity to anticodon nucleases (*Winther et al., 2016*; *Cruz et al., 2015*; *Chauhan et al., 2022*), we compared them to those of their host. We found that all 10 mycobacteriophage-encoded tRNAs that are targeted by anticodon nucleases contained anticodon loop mutations known to affect cleavage by VapC (*Figure 1C*, *Supplementary file 1a*, and *Supplementary file 1b*). These findings support the idea that phage-encoded tRNAs are insensitive to cleavage by anticodon nucleases (*Figure 1C*). We propose that these phage tRNAs serve as a mechanism to counteract the depletion of tRNAs by anticodon nucleases during phage infection, thereby allowing the phage to translate its proteins and successfully complete the infection cycle (*Figure 1D*).

In addition, we observe that mycobacteriophages rarely encode tRNAs that are cleaved within the anticodon itself (*Supplementary file 1b*), suggesting that these anticodons are avoided by the phage and that no tRNAs evolved to be insensitive to cleavage. Specifically, this avoidance is seen for the serine-coding tRNAs that are cleaved at the GA site within the anticodon (tRNA-Ser(gga), tRNA-Ser(tga), tRNA-Ser(cga), and tRNA-Ser(aga)) (*Winther et al., 2016*; *Supplementary file 1b*). To compensate for this, the phage encodes a serine tRNA (tRNA-Ser(gct)) that is not targeted by nucleases. We observed the same avoidance for known targets of RelE (*Pedersen et al., 2003*), including stop codon (cta), tRNA-Leu(tta), and tRNA-Gln(cga) (*Supplementary file 1b*). Interestingly, when comparing the codon frequency of phage genes, we observed no differences in the codon frequency between codons for which the phage encodes a tRNA and those for which it does not (Welch's two-sample $t$-test, $t = 1.0471$, df = 41.591, p-value=0.3011) (*Supplementary file 1b*). Moreover, we did not observe a difference in the codon frequency between phage and host genes (paired $t$-test, p-value=0.999) (*Supplementary file 1b*). We also found that in only 2 out of 23 instances the preferred codon (i.e. the most frequently encoded codon per amino acid) of the phage did not match that of the host (tRNA-Val(cac) and tRNA-Ala(cgc)) (*Supplementary file 1b*). Together, these

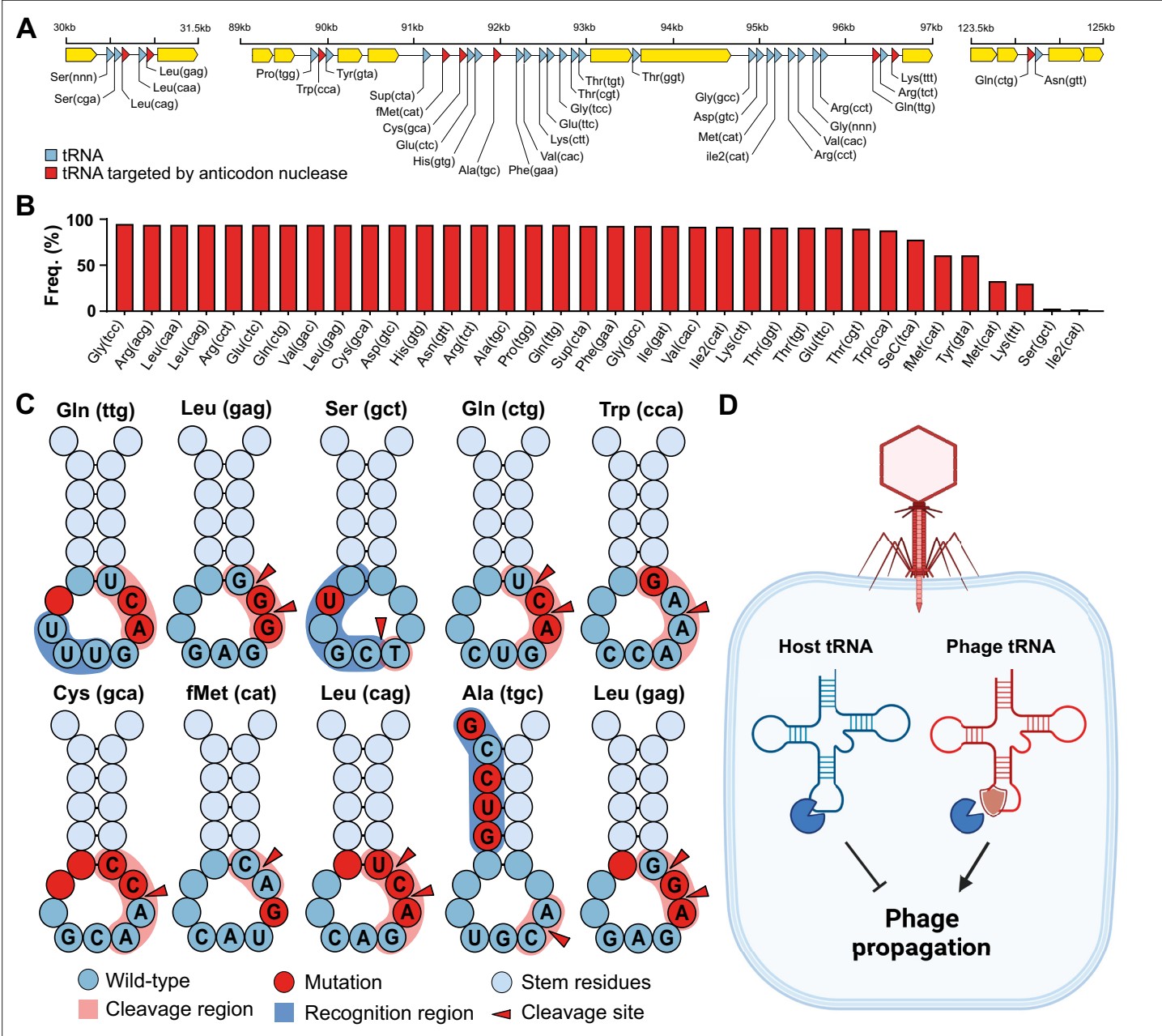

**Figure 1.** Phage transfer RNAs (tRNAs) are predicted to be anticodon nuclease resistant. (**A**) The genomic context of the tRNA clusters containing 36 tRNAs present in C1 mycobacteriophage Rizal (***Russell and Hatfull, 2017***). (**B**) Prevalence of individual phage-encoded tRNAs in the C1 mycobacteriophage cluster, composed of 161 phages. (**C**) Mutations in the anticodon loop of phage tRNAs in comparison to host tRNAs, located in the cleavage site of anticodon nucleases. (**D**) Proposed mechanism of action of phage tRNAs. During phage infection, tRNAses are activated and deplete the host tRNA pool via tRNA cleavage to prevent phage propagation. Phage tRNAs are insensitive to cleavage, allowing the phage to propagate.

The online version of this article includes the following figure supplement(s) for figure 1:

**Figure supplement 1.** Phage transfer RNAs (tRNAs) from enterobacteria.

observations suggest that the phage-encoded tRNAs were likely not selected for codon compensation. Overall, our findings support the hypothesis that phage tRNAs in mycobacteria evolved to resist anticodon nucleases in order to overcome host tRNA-depletion strategies that limit phage propagation. To investigate whether this hypothesis could extend more generally outside mycobacteria, we examined other species with anticodon nucleases, including *Shigella flexneri* (VapC-LT2) (***Winther and Gerdes, 2011***), *Escherichia coli* (VapC, PrrC, Colicin D, and Colicin E5) (***Winther and Gerdes,***

*2011*; *Kaufmann, 2011*; *Amitsur et al., 1987*; *Ogawa et al., 2006*), and *Salmonella enterica* (VapC/MvpT) (*Winther and Gerdes, 2011*). It is important to point out that the following analyses are based solely on the known cleavage site, and that the recognition sequence of these anticodon nucleases is unknown, thus limiting the ability to identify possible insensitivity-causing mutations in the recognition site of the anticodon loop. Despite this limitation, we observed mutations in the anticodon loop near or at the cleavage site of anticodon loop-targeted phage-encoded tRNAs in nearly all instances (9 out of 11) (*Figure 1—figure supplement 1* and *Supplementary file 1c*). When the anticodon itself was the target of the anticodon nuclease, as is the case for VapC-LT2 in *S. flexneri*, we found that the phage generally avoided encoding the target tRNA (tRNA-fMet), except for two instances. One of these is the phage-encoded tRNA-Lys(ttt) in coliphages. In this instance, the phage encodes a tRNA-Lys ligase (rnl1 and rnl2) that repairs tRNA-Lys(ttt) after it has been cleaved by PrrC (*Kaufmann, 2011*). The other instance involves phage-encoded tRNAs targeted by *E. coli* Colicin E5. The cleavage activity of Colicin E5 depends on modifying the wobble position (*Ogawa et al., 2006*), which might be absent in these phage tRNAs possibly caused by mutations in the anticodon loop that we observed for three out of the four targeted tRNAs (*Figure 1—figure supplement 1c*). Alternatively, these mutations might affect the recognition by Colicin E5. In summary, our findings in mycobacteria are consistent with almost all (9 out of 11) currently known instances of targeted phage tRNAs in enterobacteria (*Supplementary file 1c*). The discrepancies in these cases can be accounted for by the presence of tRNA ligases, reliance on tRNA modifications of the cleavage site, or mutations within the uncharacterized recognition site. Thus, we speculate that our hypothesis may be extended beyond mycobacteria and enterobacteria, given the ubiquity of virus-encoded tRNAs and host tRNAses (*Ogawa et al., 2006*; *Cavard and Lazdunski, 1979*; *Jones et al., 2017*).

## Conclusion

We propose that phage-encoded tRNAs escape targeting by host tRNAses via insensitivity-causing mutations within the tRNA cleavage and recognition site. This proposed hypothesis can be helpful in selecting or engineering bacteriophages capable of infecting hosts containing anticodon nucleases.

## Materials and methods

### tRNA analysis in mycobacteria and mycobacteriophages

*Mycobacterium smegmatis* MC$^2$-155 (CP000480.1) and *Mycobacterium tuberculosis* H37Rv (NC_000962.3) were used as references for obtaining the host tRNA sequences. All C1 cluster mycobacteriophage genomes were downloaded from https://phagesdb.org/ on September 1, 2022. tRNAs were annotated using Aragorn (v1.2.41; *Laslett and Canback, 2004*) and tRNAscan-SE (v2.0; *Chan and Lowe, 2019*). Codon frequency and fraction were determined using cusp (EMBOSS v6.6.0.0; *Rice et al., 2000*).

### tRNA analysis in non-mycobacteria species and phages

We performed a literature search for all anticodon nucleases with known specificities on March 1, 2023. These were found in *Leptospira interrogans* (*Lopes et al., 2014*), *S. flexneri* (*Winther and Gerdes, 2011*), *E. coli* (*Winther and Gerdes, 2011*; *Kaufmann, 2011*; *Amitsur et al., 1987*; *Ogawa et al., 2006*), *S. enterica* (*Winther and Gerdes, 2011*), *Deinococcus radiodurans* (*Miyamoto et al., 2017*), and *Geobacillus kaustophilus* (*Davidov and Kaufmann, 2008*). We excluded *L. interrogans*, *D. radiodurans*, and *G. kaustophilus* due to the absence of known tRNA-encoding phages. The species-specific tRNA sequence of the host for each anticodon in question was obtained using tRNAviz (*Lin et al., 2019*). Species-specific phages were obtained from the public PhageAI database (*Tynecki et al., 2020*) on March 1, 2023. tRNAs were annotated and analyzed as indicated above for mycobacteria and mycobacteriophages.

## Acknowledgements

This work was supported by grants from the European Research Council CoG under the European Union's Horizon 2020 research and innovation program (grant agreement no. 101003229) and the Netherlands Organisation for Scientific Research (NWO VICI; VI.C.192.027) to SJJB. We would like to thank Dr. Wim de Leeuw and Dr. Han Rauwerda (University of Amsterdam, Swammerdam Institute for

Life Sciences, MAD/RB&AB) for the use of the Crunchomics computer cluster. We thank Prof. Maria Suarez Diez and Arno Hagenbeek from Wageningen University & Research for their early contributions to this work.

## Additional information

### Funding

| Funder | Grant reference number | Author |
|---|---|---|
| European Research Council | 101003229 | Daan F van den Berg Stan JJ Brouns |
| Nederlandse Organisatie voor Wetenschappelijk Onderzoek | VI.C.192.027 | Stan JJ Brouns Ana Rita Costa |

The funders had no role in study design, data collection and interpretation, or the decision to submit the work for publication.

### Author contributions

Daan F van den Berg, Conceptualization, Data curation, Software, Formal analysis, Investigation, Methodology, Writing - original draft; Baltus A van der Steen, Investigation; Ana Rita Costa, Supervision, Investigation; Stan JJ Brouns, Supervision, Funding acquisition, Investigation, Project administration, Writing - review and editing

### Author ORCIDs

Daan F van den Berg http://orcid.org/0000-0002-2217-4074
Baltus A van der Steen http://orcid.org/0000-0001-8193-4814
Ana Rita Costa http://orcid.org/0000-0001-6749-6408
Stan JJ Brouns http://orcid.org/0000-0002-9573-1724

### Decision letter and Author response

Decision letter https://doi.org/10.7554/eLife.85183.sa1
Author response https://doi.org/10.7554/eLife.85183.sa2

## Additional files

### Supplementary files

• Supplementary file 1. Phage versus host-encoded transfer RNA (tRNA) comparisons. (**a**) Overview of all C1 mycobacteriophage encoded tRNAs. (**b**) Overview of anticodons encoded by C1 mycobacteriophages compared to their hosts'. (**c**) Analysis of anticodon loop mutations and anticodon avoidance in other species.

• MDAR checklist

### Data availability

An overview of the analysed data supporting the findings of this study are available within the paper and in the Supplementary Data. All genomic sequences of the C1 mycobacteriophages were obtained from the publicly available actinobacteriophage database (PhagesDB; link: https://phagesdb.org/subclusters/C1/). Mycobacterium smegmatis MC2-155 (CP000480.1) and Mycobacterium tuberculosis H37Rv (NC_000962.3) were used as the reference for obtaining the host tRNA sequences.

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
