## [Editor Report]

This important work substantially advances our understanding of the mechanisms phages use to evade host defenses. Specifically, the authors use computational and theoretical analyses of tRNA-encoding phages that infect several bacterial species to identify a novel, potential mechanism through which phage-encoded tRNAs help these phages evade tRNA cleavage that is induced as a host defense. Although the evidence supporting the conclusions is compelling, with multiple observations suggesting that the phage-encoded tRNAs have evolved to evade host-encoded tRNases, the conclusions would have been more strongly supported by providing an experimental test of the hypothesis.

---

## [Decision Letter]

**Decision letter after peer review:**

Thank you for submitting your article "Phage tRNAs evade tRNA-targeting host defenses through anticodon loop mutations" for consideration by *eLife*. Your article has been reviewed by 3 peer reviewers, one of whom is a member of our Board of Reviewing Editors, and the evaluation has been overseen by James Manley as the Senior Editor. The following individual involved in the review of your submission has agreed to reveal their identity: Blake Wiedenheft (Reviewer #3).

Essential revisions:

1) The scope of the analysis is currently not in line with the claims made. Thus, the authors must either extend the analysis to more phages and hosts or soften the claims. At a minimum, the authors must explain the rationale for limiting the analysis to a single phage-host group, explain/justify why the scope of their analysis is so narrow, and modify the claims in the manuscript so that the limitations of the work are more clear.

2) The authors must respond to each of the 8 points listed under Reviewer #2's Recommendations for the Authors, below. They should pay particular attention to Point 7. Perhaps the sentence simply needs clarification, but, as written, it seems to present some contradiction. The authors say "selection of phage tRNAs is only determined by their insensitivity to tRNAses", but in that case how can it be that "phage genes do not avoid codons of cleaved tRNAs"? It could be that the codons of cleaved tRNAs are also found in phage-encoded mutated tRNAs, but then why would we expect the phages to avoid them? Perhaps a better analysis would be to compare the anticodons of phage-encoded and host-encoded tRNAs. If they are broadly similar, that would mean that the phages are not encoding their own tRNA to complement the pool of tRNA with anticodons that the host does not provide.

*Reviewer #1:*

In this computational and theoretical study, van den Berg et al. present a new hypothesis for why many classes of phages encode a large number of tRNAs. Specifically, they propose that phage-encoded tRNAs are resistant to host-encoded tRNases that are activated by the host in response to phage infection. Consequently, phage-encoded tRNAs translation and phage propagation proceed despite the tRNA degradation induced by the tRNases. To investigate this hypothesis, the authors perform a sequence analysis of one class of phages that affect mycobacteria. They demonstrate that, of the 10 phage-encoded tRNAs that are expected to be targeted by mycobacterial tRNases, they all exhibit mutations in the tRNase recognition or cleavage sites that would be expected to prevent tRNA cleavage. In addition, the authors provide two examples supporting the idea that the sequences of phage-encoded tRNAs have evolved under selective pressure to avoid sequences expected to be cleaved by mycobacterial tRNases. Finally, the authors show that phage-encoded genes do not seem to have evolved under selective pressure to avoid codons read by tRNAs that are targeted by mycobacterial tRNases and do not seem to have a preference for codons read by tRNAs that are resistant to these tRNases.

A major strength of this work is that the computational and theoretical findings described above strongly support the authors' hypothesis. A major weakness, however, is that the analysis was performed on one phage class that affects one bacterial genus. It is therefore not clear how widely these findings hold across different phage classes and bacterial genera. Another weakness is the lack of an experimental test of the authors' hypothesis.

Despite the weaknesses, I expect that this work will have a significant impact on the field. This is largely due to the recent explosion of interest in the mechanisms through which phages avoid host defenses (e.g., CRISPR, CRISPR-associated transposition, etc.). This is therefore very timely work. In addition, I imagine that the computational and theoretical work presented here will prompt experimentalists to test the hypothesis presented by the authors.

The work presented here would be greatly strengthened by expanding the computational and theoretical analysis to include additional phage classes and bacterial genera. Otherwise, it is unclear how widespread the findings are in phage/bacterial biology. It should be possible and relatively easy for the authors to expand their analyses.

The work presented here would be even more greatly strengthened by an experimental test of the computationally and theoretically well-supported hypothesis presented here. Nonetheless, I understand that such an experimental test is well beyond the scope of the current work.

*Reviewer #2:*

van De Berg et al. report on the specific tRNAs encoded in a group of mycobacteriophage, and how they differ from the tRNAs encoded in two host genomes. They further argue that these variations can be explained (and are probably driven) by differences in sensitivity to tRNAses. The hypothesis and observations are interesting, although the manuscript only reports on a narrow group of hosts and their (closely related) phages. It does suggest however that tRNA degradation as a defense mechanism should be further investigated to understand how widespread this phenomenon is, how much it accounts for the presence of tRNAs in phage genomes, and how it may impact codon usage or translation efficiency at different stages of phage infection.

1. General comments: Please make sure to include line numbers in your manuscript to help with the review process. Please also make sure that the manuscript clearly distinguishes between new results, new analyses that confirmed previous/published results, and previous/published results that are provided for context to the reader.

2. Abstract: "bacterial genera" may be more clear as "bacterial host genera"?

3. "Based on mutational patterns of tRNA anticodon loops, we predict that phage tRNAs are insensitive to the host tRNAses": I was surprised to only find only a relatively simple analysis of a single host-phage group to support this broad claim. As currently written, this manuscript reads to me more like a nice preliminary observation rather than a complete study: there is no experimental confirmation of the hypothesis, the analysis is not extended beyond a single phage group and a single host, and the only statistical test performed was run on whole phage genomes when, most likely, the pattern the authors would expect would be specific to some (group of) genes only.

4. Introduction:

The current introduction is a bit short in my opinion, and could benefit from being expanded on some of the following topics: at which stage in the infection cycle do the depletion of host tRNAs occur, and how is it triggered? Is it more a defense used against lytic or temperate phages, and why? What is the mechanism(s) that hosts use to deplete the tRNA pool? Note that some information is currently in the first section of the results ("We hypothesize […] Figure 1D."), which I would argue would be a better fit in the introduction section instead.

5. Results and discussion:

"We compared the tRNAs encoded by phages with those of their host and observed all 10 phage-encoded tRNAs that are known to be targeted by anticodon nucleases to contain anticodon loop mutations (Winther et al., 2016; Cruz et al., 2015; Chauhan et al., 2022), reinforcing the idea that phage-encoded tRNAs are likely insensitive to cleavage (Figure 1C).": This sentence is unclear to me, was this a new analysis run by the authors? If so, the analysis itself should be better described before reporting the results. Otherwise, it must be made clear to the readers that this is a previously published result, e.g. "Previous studies compared.…" or "In previous studies, we compared.…"

6. "we also observed a strong counter-selection for tRNAs that are cleaved in the anticodon itself (Table S1)": Table S1 lists 168 tRNA, but it's unclear to me how a reader can go from this list to an observation of counter-selection for tRNAs predicted to be cleaved? I also could not find a legend to Table S1, so I don't understand some of the columns ("Manual ACL to host" ?), or even what the "Sequence_ID" refer to (phage genes ? Host genes ?)

7. "we observed that phage genes do not avoid codons of cleaved tRNAs, nor do they have a preference for codons with nuclease insensitivity (Welch Two Sample t-test, t = 0.53848, df = 41.583, p-value > 0.05), suggesting that the selection of phage tRNAs is only determined by their insensitivity to tRNAses and not by codon usage.": I am not sure I understand this claim. The way I understand it, either the phage tRNAs recognize the same codon as the host tRNAs, in which case there is no reason to think that codon usage would be impacted (regardless of tRNAse activity), or the phage and host tRNAs recognize different codons, in which case a selection for nuclease insensitivity should impact codon usage? I am not sure what I am missing here. Could the authors clarify how they link this lack of codon usage bias they observe with a selection determined by insensitivity?

8. Material and Methods: This section should provide more information on how the statistical enrichment/depletion in codon was performed.

*Reviewer #3:*

In this short paper, van den Berg et al. hypothesize that viral tRNAs are resistant to host-encoded nucleases that suppress viral infections by destroying tRNAs. To support this hypothesis, they compared the tRNAs encoded in 161 mycobacteriophage genomes with those of their host and show that phage tRNAs contain mutations anticipated to make them resistant to cleavage by host nucleases. While the paper is unusually short and lacks biochemical evidence supporting the sequence observations, the sequence comparisons are compelling, and the concept has broad implications.

Figure 1a highlights the genomic context of 36 tRNAs present in C1 mycobacteriophage Rizal. A subset of these is anticipated to be targeted host anticodon nucleases (red). Consider comment on the other tRNAs and their biological roles.

---

## [Author Response]

Essential revisions:1) The scope of the analysis is currently not in line with the claims made. Thus, the authors must either extend the analysis to more phages and hosts or soften the claims. At a minimum, the authors must explain the rationale for limiting the analysis to a single phage-host group, explain/justify why the scope of their analysis is so narrow, and modify the claims in the manuscript so that the limitations of the work are more clear.

To explore our hypothesis that phage tRNAs counteract bacterial host anticodon nucleases, we required hosts with well-studied anticodon nucleases and a large set of infecting phages with tRNAs. For this reason, we chose to investigate our hypothesis in Mycobacterium species, which has the most extensively researched (and diverse) anticodon nucleases with known target specificity. To broaden our observations, we expanded our search for all anticodon nucleases with known target specificity and found several additional options to explore, including *E. coli* (VapC, PrrC, colicin D, and colicin E5), *S. flexneri* (VapC-LT2), *L. interrogans* (VapC-L.interrogans), and *S. typhimurium* (VapC/MvpT). Although the datasets for these species are more limited, we observe the same pattern of resistance mutations and anticodon avoidance in these species (Supplementary File 1b and Figure 1-Supplementary Figure 1a). Although we have extended our observations to other species, we have also revised our text to soften our claims where appropriate.

2) The authors must respond to each of the 8 points listed under Reviewer #2's Recommendations for the Authors, below. They should pay particular attention to Point 7. Perhaps the sentence simply needs clarification, but, as written, it seems to present some contradiction. The authors say "selection of phage tRNAs is only determined by their insensitivity to tRNAses", but in that case how can it be that "phage genes do not avoid codons of cleaved tRNAs"? It could be that the codons of cleaved tRNAs are also found in phage-encoded mutated tRNAs, but then why would we expect the phages to avoid them? Perhaps a better analysis would be to compare the anticodons of phage-encoded and host-encoded tRNAs. If they are broadly similar, that would mean that the phages are not encoding their own tRNA to complement the pool of tRNA with anticodons that the host does not provide.

We have addressed all 8 points raised by Reviewer #2 in detail below and clarified the text throughout. As for point 7, we have performed the suggested comparison between the anticodons of the phage-encoded and host-encoded tRNAs and found that 29 out of 30 phage-encoded anticodons are also encoded by the host (Supplementary File 1b). In addition, we have compared the codon frequency of the phage and host and found no difference (Paired t-test, p-value = 0.999). Moreover, we found that in only 2 out of 23 instances, the preferred codon (i.e., the most frequently encoded codon per amino acid) of the phage did not match that of the host. Together, these observations suggest that the phage-encoded tRNAs were likely not selected for codon compensation.

Reviewer #1:In this computational and theoretical study, van den Berg et al. present a new hypothesis for why many classes of phages encode a large number of tRNAs. Specifically, they propose that phage-encoded tRNAs are resistant to host-encoded tRNases that are activated by the host in response to phage infection. Consequently, phage-encoded tRNAs translation and phage propagation proceed despite the tRNA degradation induced by the tRNases. To investigate this hypothesis, the authors perform a sequence analysis of one class of phages that affect mycobacteria. They demonstrate that, of the 10 phage-encoded tRNAs that are expected to be targeted by mycobacterial tRNases, they all exhibit mutations in the tRNase recognition or cleavage sites that would be expected to prevent tRNA cleavage. In addition, the authors provide two examples supporting the idea that the sequences of phage-encoded tRNAs have evolved under selective pressure to avoid sequences expected to be cleaved by mycobacterial tRNases. Finally, the authors show that phage-encoded genes do not seem to have evolved under selective pressure to avoid codons read by tRNAs that are targeted by mycobacterial tRNases and do not seem to have a preference for codons read by tRNAs that are resistant to these tRNases.A major strength of this work is that the computational and theoretical findings described above strongly support the authors' hypothesis. A major weakness, however, is that the analysis was performed on one phage class that affects one bacterial genus. It is therefore not clear how widely these findings hold across different phage classes and bacterial genera. Another weakness is the lack of an experimental test of the authors' hypothesis.Despite the weaknesses, I expect that this work will have a significant impact on the field. This is largely due to the recent explosion of interest in the mechanisms through which phages avoid host defenses (e.g., CRISPR, CRISPR-associated transposition, etc.). This is therefore very timely work. In addition, I imagine that the computational and theoretical work presented here will prompt experimentalists to test the hypothesis presented by the authors.

We thank the reviewer for the encouraging review of our manuscript. We have addressed the reviewer’s concern about the limited analysis by expanding our investigation to include other Enterobacteria species with anticodon nucleases of known target specificity. These species are *E. coli* (VapC, PrrC, Colicin D, and Colicin E5), *S. flexneri* (VapC-LT2), *L. interrogans* (VapC-L.interrogans), and *S. typhimurium* (VapC/MvpT). Although the datasets for these species are more limited than for *M. tuberculosis*, our findings remain consistent with the same pattern of resistance mutations and anticodon avoidance observed in mycobacteriophages (Supplementary File 1b). We also revised our language to be more cautious regarding our claims, including acknowledging the lack of wet lab validation.

The work presented here would be greatly strengthened by expanding the computational and theoretical analysis to include additional phage classes and bacterial genera. Otherwise, it is unclear how widespread the findings are in phage/bacterial biology. It should be possible and relatively easy for the authors to expand their analyses.The work presented here would be even more greatly strengthened by an experimental test of the computationally and theoretically well-supported hypothesis presented here. Nonetheless, I understand that such an experimental test is well beyond the scope of the current work.

We appreciate the reviewer’s positive comments and constructive criticism of our manuscript. We limited ourselves to phages infecting Mycobacteria due to the abundance of information available on tRNAses (e.g. VapCs) and their targets in this bacterial host. To broaden our observations/claims, we now also include an analysis performed on all characterized anticodon nucleases, including *E. coli* (VapC, PrrC, Colicin D, and Colicin E5), *S. flexneri* (VapC-LT2), *L. interrogans* (VapC-L.interrogans), and *S. typhimurium* (VapC/MvpT). Although the datasets for these species are limited, we observe the same pattern of resistance mutations and anticodon avoidance (Supplementary File 1c). We appreciate the reviewer’s suggestion to experimentally validate our hypothesis, which would indeed provide valuable insights into the mechanisms of phage tRNA resistance to host-encoded tRNA nuclease. We hope that our study will motivate further investigations and collaborations to explore these mechanisms in more detail.

Reviewer #2:van De Berg et al. report on the specific tRNAs encoded in a group of mycobacteriophage, and how they differ from the tRNAs encoded in two host genomes. They further argue that these variations can be explained (and are probably driven) by differences in sensitivity to tRNAses. The hypothesis and observations are interesting, although the manuscript only reports on a narrow group of hosts and their (closely related) phages. It does suggest however that tRNA degradation as a defense mechanism should be further investigated to understand how widespread this phenomenon is, how much it accounts for the presence of tRNAs in phage genomes, and how it may impact codon usage or translation efficiency at different stages of phage infection.

We appreciate the reviewer’s positive feedback on our manuscript. We have expanded our analysis to include other species with anticodon nucleases of known target specificity, and our findings remain consistent with the same pattern of resistance mutations and anticodon avoidance observed in mycobacteriophages (Supplementary File 1b).

1. General comments: Please make sure to include line numbers in your manuscript to help with the review process. Please also make sure that the manuscript clearly distinguishes between new results, new analyses that confirmed previous/published results, and previous/published results that are provided for context to the reader.

In the revised version, line numbers have been added and the text has been edited to clearly distinguish between new results, re-confirmed previous results through new analysis, and published results provided for context.

2. Abstract: "bacterial genera" may be more clear as "bacterial host genera"?

Modified as suggested.

3. "Based on mutational patterns of tRNA anticodon loops, we predict that phage tRNAs are insensitive to the host tRNAses": I was surprised to only find only a relatively simple analysis of a single host-phage group to support this broad claim. As currently written, this manuscript reads to me more like a nice preliminary observation rather than a complete study: there is no experimental confirmation of the hypothesis, the analysis is not extended beyond a single phage group and a single host, and the only statistical test performed was run on whole phage genomes when, most likely, the pattern the authors would expect would be specific to some (group of) genes only.

To address the reviewer’s concerns, we have expanded our bioinformatic analysis to include all species that contain characterized anticodon nucleases with known specificities, including *E. coli* (VapC, PrrC, colicin D, and colicin E5), *S. flexneri* (VapC-LT2), *L. interrogans* (VapC-L.interrogans), and *S. typhimurium* (VapC/MvpT). Although the datasets for these species are more limited, we observe the same pattern of resistance mutations and anticodon avoidance that we described in mycobacteriophages (Supplementary File 1c). Furthermore, we also revised our language to be more cautious regarding our claims, including acknowledging the lack of wet lab validation.

4. Introduction:The current introduction is a bit short in my opinion, and could benefit from being expanded on some of the following topics: at which stage in the infection cycle do the depletion of host tRNAs occur, and how is it triggered? Is it more a defense used against lytic or temperate phages, and why? What is the mechanism(s) that hosts use to deplete the tRNA pool? Note that some information is currently in the first section of the results ("We hypothesize […] Figure 1D."), which I would argue would be a better fit in the introduction section instead.

As suggested, the introduction of the manuscript has been expanded to include answers to the questions posed. We have also moved the current first section of the results to the introduction.

1) At which stage in the infection cycle does the depletion of host tRNAs occur?

Line 30:

“A recent study by Yang et al. (2021) may have hinted at an additional factor: phage tRNAs represent a means to counteract the rapid depletion of host tRNAs that occurs as an early response to phage infection (Thompson and Parker, 2009; Yang et al., 2021; Jain et al., 2021; Amitsur et al., 1989; Wolfram-Schauerte et al., 2022).”

2) and how is it triggered?

Line 36:

“Exactly what activates host tRNAs is often unknown except for the anticodon nuclease PrrC, which cleaves tRNA-Lys(ttt) when it is triggered after sensing phage-encoded protein Stp (Kaufmann, 2011).”

3) Is it more a defense used against lytic or temperate phages, and why?

Line 25:

“During the 1960s, tRNAs were also reported in viruses of bacteria (phages) (Weiss et al., 1968). We now know that phage-encoded tRNAs are widespread, especially in virulent phages (Bailley-Bechet et al., 2007).”

4) What are the mechanism(s) that hosts use to deplete the tRNA pool?

Line 34:

“The host uses several mechanisms to deplete its tRNA pool, such as down-regulating the expression of its tRNAs, modifying them to make them unusable for translation, and most notably cleaving the tRNAs using anticodon nucleases.”

5) Note that some information is currently in the first section of the results ("We hypothesize […] Figure 1D)."), which I would argue would be a better fit in the introduction section instead.

We have moved this part to the introduction section.

We also added information regarding a recent preprint from Azam et al. (2023) on line 39:

“Recently, a phage encoding a tRNA was found to replenish the host tRNA depleted by the Retron Ec78 phage defense system, thereby preventing the inhibition of phage propagation (Azam et al., 2023).”

5. Results and discussion:"We compared the tRNAs encoded by phages with those of their host and observed all 10 phage-encoded tRNAs that are known to be targeted by anticodon nucleases to contain anticodon loop mutations (Winther et al., 2016; Cruz et al., 2015; Chauhan et al., 2022), reinforcing the idea that phage-encoded tRNAs are likely insensitive to cleavage (Figure 1C).": This sentence is unclear to me, was this a new analysis run by the authors? If so, the analysis itself should be better described before reporting the results. Otherwise, it must be made clear to the readers that this is a previously published result, e.g. "Previous studies compared.…" or "In previous studies, we compared.…"

We revised the text to better distinguish between previous results and results from analyses performed in this study.

“To check phage tRNAs for mutations that are known to cause insensitivity to anticodon nucleases (Winther et al., 2016; Cruz et al., 2015; Chauhan et al., 2022), we compared them to those of their host. We found that all 10 phage-encoded tRNAs that are known to be targeted by anticodon nucleases contained anticodon loop mutations known to affect cleavage (Supplementary File 1a and Supplementary File 1b). These findings support the idea that phage-encoded tRNAs are insensitive to cleavage by anticodon nucleases (Figure 1C).”

6. "we also observed a strong counter-selection for tRNAs that are cleaved in the anticodon itself (Table S1)": Table S1 lists 168 tRNA, but it's unclear to me how a reader can go from this list to an observation of counter-selection for tRNAs predicted to be cleaved? I also could not find a legend to Table S1, so I don't understand some of the columns ("Manual ACL to host" ?), or even what the "Sequence_ID" refer to (phage genes ? Host genes ?)

We have added a description to Supplementary Figure 1a and improved the descriptions of the columns. We added an abundance column that shows the abundance in percentages of C1 mycobacteriophages that carry specific tRNA sequences. We also removed columns that were not relevant to this work (tRNA_tail, host_tail, tRNA Type, and Accep. Host). Lastly, we created Supplementary Figure 1b to show the anticodons encoded by the host and phage.

7. "we observed that phage genes do not avoid codons of cleaved tRNAs, nor do they have a preference for codons with nuclease insensitivity (Welch Two Sample t-test, t = 0.53848, df = 41.583, p-value > 0.05), suggesting that the selection of phage tRNAs is only determined by their insensitivity to tRNAses and not by codon usage.": I am not sure I understand this claim. The way I understand it, either the phage tRNAs recognize the same codon as the host tRNAs, in which case there is no reason to think that codon usage would be impacted (regardless of tRNAse activity), or the phage and host tRNAs recognize different codons, in which case a selection for nuclease insensitivity should impact codon usage? I am not sure what I am missing here. Could the authors clarify how they link this lack of codon usage bias they observe with a selection determined by insensitivity?

We thank the reviewer for bringing up this point. To clarify the wording, we have changed “codon usage” to “codon frequency”. When we compared to codon frequency of the phage to its host, we observed no difference (paired t-test, p = 0.999) (Supplementary Figure 1b). Additionally, we analyzed if frequently used codons by the phage, but rarely used by the host, are encoded by the phage to supply the tRNA pool of rare anticodons. To check this, we looked for codons that are preferred by the phage but rarely used by the host. In only 2 instances (tRNA-Val(cac) and tRNA-Ala(cgc)) out of 23 preferred codons (the most frequently encoded codon per amino acid) (Supplementary Figure 1b) that correspond to anticodons encoded by the phage, there was a difference between the phage and host-preferred codon, further supporting the notion that codon compensation is not a factor. Moreover, we have performed a comparison between the anticodons of the phage-encoded and host-encoded tRNAs and found that 29 out of 30 phage-encoded anticodons are also encoded by the host (Supplementary Figure 1b). We clarified the text as follows:

“Interestingly, when comparing the codon frequency of phage genes, we observed no differences in the codon frequency between codons for which the phage encodes a tRNA and those for which it does not (Welch Two Sample t-test, t = 1.0471, df = 41.591, p-value = 0.3011) (Supplementary Figure 1b). Moreover, we did not observe a difference in the codon frequency between phage and host genes (Paired t-test, p-value = 0.999) (Supplementary Figure 1b). We also found that in only 2 out of 23 instances, the preferred codon (i.e. the most frequently encoded codon per amino acid) of the phage did not match that of the host (tRNA-Val(cac) and tRNA-Ala(cgc)) (Supplementary Figure 1b). Together, these observations suggest that the phage-encoded tRNAs were likely not selected for codon compensation.”

8. Material and Methods: This section should provide more information on how the statistical enrichment/depletion in codon was performed.

Our Material and Methods section has been carefully reviewed and updated to reflect the changes made to the manuscript.

Reviewer #3:In this short paper, van den Berg et al. hypothesize that viral tRNAs are resistant to host-encoded nucleases that suppress viral infections by destroying tRNAs. To support this hypothesis, they compared the tRNAs encoded in 161 mycobacteriophage genomes with those of their host and show that phage tRNAs contain mutations anticipated to make them resistant to cleavage by host nucleases. While the paper is unusually short and lacks biochemical evidence supporting the sequence observations, the sequence comparisons are compelling, and the concept has broad implications.

We appreciate the reviewer’s positive assessment of our manuscript. Below, we have provided detailed responses to the specific comments.

Figure 1a highlights the genomic context of 36 tRNAs present in C1 mycobacteriophage Rizal. A subset of these is anticipated to be targeted host anticodon nucleases (red). Consider comment on the other tRNAs and their biological roles.

We suspect that undiscovered host anticodon nucleases may also target other phage-encoded tRNAs, as evidenced by similar mutations in these tRNAs (Supplementary Figure 1a). It should be noted that these mycobacteriophages encode a large portion of the host’s codons (29 out of 50) (Supplementary Figure 1b), suggesting that the primary purpose might be to replace a large part of the tRNA pool of the host, without any specific biological role apart from replacement. This aligns with the observation that some phages downregulate the host tRNAs to completely take over the tRNA pool (Son et al., 2022). Additionally, it is possible that some of these other tRNAs serve distinct biological functions, such as regulation. For example, some tRNAs may be intentionally encoded for cleavage, which has been shown in mammals to have distinct regulatory functions, such as interference RNAs (Fazio et al., 2022). Lastly, a recent preprint by Azam et al. (2023) sheds new light on the possibility that phage-encoded tRNAs are specific to replenish certain cleaved tRNAs targeted by the host anti-phage defense system. This is a more likely role for the tRNAs encoded by phages with few tRNAs, which supports our findings. However, the above-mentioned hypotheses for the role of these other phage-encoded tRNAs are too speculative and were difficult to add to the paper without deviating too much from the primary storyline.